

# ERISNet: deep neural network for *Sargassum* detection along the coastline of the Mexican Caribbean

Javier Arellano-Verdejo[1,*], Hugo E. Lazcano-Hernandez[2,*] and Nancy Cabanillas-Terán[2]

[1] Estacion para la Recepcion de Informacion Satelital ERIS-Chetumal, El Colegio de la Frontera Sur, Chetumal, Quintana Roo, México
[2] Catedras CONACYT-El Colegio de la Frontera Sur, Chetumal, Quintana Roo, México
[*] These authors contributed equally to this work.

## ABSTRACT

Recently, Caribbean coasts have experienced atypical massive arrivals of pelagic *Sargassum* with negative consequences both ecologically and economically. Based on deep learning techniques, this study proposes a novel algorithm for floating and accumulated pelagic *Sargassum* detection along the coastline of Quintana Roo, Mexico. Using convolutional and recurrent neural networks architectures, a deep neural network (named ERISNet) was designed specifically to detect these macroalgae along the coastline through remote sensing support. A new dataset which includes pixel values with and without *Sargassum* was built to train and test ERISNet. Aqua-MODIS imagery was used to build the dataset. After the learning process, the designed algorithm achieves a 90% of probability in its classification skills. ERISNet provides a novel insight to detect accurately algal blooms arrivals.

## INTRODUCTION

Pelagic *Sargassum* is formed by brown macroalgae *S. fluitans* and *S. natans*, and constitutes floating ecosystems serving as habitats and nurseries for important marine species like sea turtles, fishes, invertebrates, and micro and macro-epiphytes (*Rooker, Turner & Holt, 2006*; *Witherington, Hirama & Hardy, 2012*). However, over the last seven years, Caribbean shores have experienced atypical massive shoals of pelagic *Sargassum*, with exceptional abundances for the Mexican Caribbean during the summers of 2015 and 2018. Massive influx was observed in numerous Caribbean beaches linked with the accumulation of *Sargassum* spp. (hereafter *Sargassum*)(*Gower, Young & King, 2013*; *van Tussenbroek et al., 2017*). Since 2011, these extensive off-shore *Sargassum* shoals have appeared in unprecedented ways in oceanic waters off the coast of northern Brazil (*De Széchy et al., 2012*; *Gower, Young & King, 2013*; *Sissini et al., 2017*), actually, this events of *Sargassum* blooms were registered on the African coast as well (*De Széchy et al., 2012*; *Maréchal, Hellio & Hu, 2017*). Those shoals likely have origins in the North Equatorial Recirculation Region (NERR)

Corresponding authors
Javier Arellano-Verdejo,
javier.arellano@mail.ecosur.mx
Nancy Cabanillas-Terán,
ncabanillas@ecosur.mx

(*Schell, Goodwin & Siuda, 2015*), suggesting that they did not emerge from the traditional northwestern Atlantic Ocean region known as "The Sargasso Sea". With MODIS (Moderate Resolution Imaging Spectroradiometer) and MERIS (Medium Resolution Imaging Spectrometer) satellite images, it was possible to track a shift in their distribution patterns in order to identify a new possible distribution source. To track the patterns of *Sargassum*, *Gower, Young & King (2013)* used the maximum chlorophyll index (MCI) derived from MERIS level 1 radiances and the MODIS Red Edge (MRE). Likewise *Sissini et al. (2017)* used MODIS Aqua 250 m level 1 radiance images (band 1, 645 nm; band 2, 859 nm) to find *Sargassum* distribution patterns.

The biomass reported since 2011 has no precedent (*van Tussenbroek et al., 2017*; *Rodríguez-Martínez, van Tussenbroek & Jordán-Dahlgren, 2016*). A crucial difference is that *Sargassum* does not remain in the open ocean, but rather washes ashore at the coast. The accumulated biomass has resulted in negative conditions both economically and ecologically (*Hu et al., 2016*; *Schell, Goodwin & Siuda, 2015*).

This excessive biomass along the coast modifies beaches and increases bioerosion. This has a direct influence on the tourist industry, and depending on the *Sargassum* amounts, these generate bad smell, disrupts the access to tourists, and even could have repercussions for health (*Maréchal, Hellio & Hu, 2017*). The accumulation is also associated with physical-chemical water change, anoxia, and generation of hydrogen sulphide (*Louime, Fortune & Gervais, 2017*). *Sargassum* sinking can contribute to the organic matter input, the shallow and the deep-sea communities as well (*Wang et al., 2018*). The decomposition of *Sargassum* biomass on the beaches is a disturbance agent that can also modify the physical, physiological and ecological processes in near-shore coral reef communities. The modified flow of organic matter caused by this disturbance could have negative effects at different scales. The negative effects would also affect tourism, local fisheries (*Cuevas, Uribe-Martínez & Liceaga-Correa, 2018*; *Ferreira et al., 2009*; *Solarin et al., 2014*), and benthic communities. The case of coral reefs is relevant, as they are the most threatened marine ecosystems in the world (*Hoegh-Guldberg et al., 2007*; *Harvey et al., 2018*) and although the whole ecological impacts remain still unknown, and we consider that one of the most affected areas of *Sargassum* accumulations is the reef lagoon. Excessive organic material leads to turbidity and reduced light causing hypoxia in seagrasses and corals. This has increased coral mortality and damaged seagrasses and associated fauna (*Franks, Johnson & Ko, 2016*; *van Tussenbroek et al., 2017*; *Louime, Fortune & Gervais, 2017*). According to *Spalding et al. (2017)* coral reefs provide nearly US$35.8 billion in net benefits of goods and services to world economies each year. This includes tourism, fisheries, and coastal protection. Caribbean region represents US$1,853 million of those benefits. Annually around 10 million tourists visit the Mexican Caribbean (*Rioja-Nieto & Álvarez-Filip, 2018*). Economic loss caused by *Sargassum* arrival can reduce those benefits. In 2015 alone, the state government invested US$3 million to remove the macroalgae from tourist areas. More than 4,400 workers were hired. In 2018, between June and August US$3.1 million were spent on wages for 450 workers.

Through spectral water-leaving radiance or surface reflectance, remote sensing has served as the primary means to study ocean constituents suspended or dissolved in water (*Dickey,*

*Lewis & Chang, 2006*). Floating Algae Index (FAI) proposed by *Hu (2009)* has been the main method used in remote sensing to assess presence/absence of floating algae in the open sea (*Hu, 2009*). Research has been carried out to assess and monitor pelagic *Sargassum* in the Western Central Atlantic, Yellow Sea, the Gulf of Mexico, and the Caribbean Sea (*Putman et al., 2018*). On the other hand, sensors with several spatial, temporal, spectral and radiometric features have been used for the study of *Sargassum* (*Dickey, Lewis & Chang, 2006*; *Hu et al., 2015*; *Cuevas, Uribe-Martínez & Liceaga-Correa, 2018*). The use of platforms Aqua-MODIS, Terra-MODIS and Landsat imagery is highlighted due to their wide coverage and worldwide heritage, as well as being open-access datasets (*Hu, 2009*; *Hu et al., 2015*; *Wang & Hu, 2016*). Regarding the coasts of the Gulf of Mexico and the Caribbean Sea, (*Cuevas, Uribe-Martínez & Liceaga-Correa, 2018*) present a methodology to detect *Sargassum* in the northeastern region of the Yucatan peninsula applying the ''Random Forest'' algorithm to a set of Landsat 8 imagery previously selected. The previous studies are valuable contributions to the detection of floating vegetation like pelagic *Sargassum*. However, no study has dealt with the probability of presence of *Sargassum* along the coastline of the Mexican Caribbean. Considering the potential damage and the negative effects of the blooms not only for the tourism, but also for the health of coastal ecosystems, it is of utmost importance to develop precise methods to detect the algal bloom events.

From the optical point of view, the oligotrophic waters of the Quintana Roo coasts are transparent under non-sargasso conditions. In 2015 and 2018 due to the constant arrival of *Sargassum*, its decomposition caused murky brown waters, which in turn altered the nearshore water surface reflectance values (*van Tussenbroek et al., 2017*).

An Artificial Neural Network (ANN) is a mathematical model inspired by the biological behavior of neurons and how they are organized. The ANNs are massive parallel systems with large numbers of interconnected simple processors. A single layer perceptron (SLP) is a feed-forward network based on a threshold transfer function. SLP is the simplest type of artificial neural network and can only classify linearly-separable cases (*Jain, Mao & Mohiuddin, 1996*). The multilayer Perceptron (MLP) is a generalization of the simple Perceptron and arose as a consequence of the limitations of said architecture in relation to the problem of non-linear separability. *Minsky & Papert (2017)* showed that the combination of several MLPs could be an adequate solution to treat certain non-linear problems. Neural networks have had many applications in various areas of knowledge such as: control systems (*Hunt et al., 1992*), business (*Vellido, Lisboa & Vaughan, 1999*), manufacturing (*Zhang & Huang, 1995*) and medicine (*Baxt, 1991*) to mention just a few.

Deep Learning (DL) was presented in Science magazine in 2006. Since then, multiple algorithms have been developed, including convolutional neural network (CNN), recurrent neural network (RNN), stacked auto-encoder (SAE) and deep belief network (DBN). Many variants of deep learning algorithms are a combination of two or more of these algorithms (*Zhang et al., 2018*).

DL is a subfield of machine learning inspired by the ANN and is formed by a set of algorithms that try to model high-level abstractions in data using architectures composed of multiple non-linear transformations (*LeCun, Bengio & Hinton, 2015*). In

DL, a Convolutional Neural Network (CNN) is a type of ANN composed of multiple layers of convolutional filters of one or more dimensions, and is very effective for tasks of artificial vision, such as classification and segmentation of images (*Schmidhuber, 2015*).

Another type of NN widely used in DL is the Recurrent Neural Network (RNN). An RNN implements a Long Short-term memory architecture (LSTM), which makes RNN an ideal tool for modeling and classifying time series (*Schmidhuber, 2015*). Deep Learning has been used successfully in multiple areas such as biomedicine (*Mamoshina et al., 2016*), medicine (*Greenspan, van Ginneken & Summers, 2016*), time series prediction (*Weigend, 2018*), speech recognition, computer vision, pattern recognition and remote sensing (*Liu et al., 2017*), among others.

Our study arose from the following hypothesis: It is possible that a Deep Neural Network (ERISNet) learns automatically the relationships among different corrected reflectances (rhot and rhos) able to detect the presence of *Sargassum* without the use of existing index (i.e., NDVI, FAI, AFAI, etc.) from a properly labeled dataset. The main objective of this study is to analyze if we can detect *Sargassum* along the Mexican Caribbean coastline by using MODIS data and Deep Learning Networks with an accuracy of more than 80%. Under this view a new algorithm for *Sargassum* detection is presented. This algorithm is based on DL techniques. Hence, our aim was to classify the presence/absence of pelagic *Sargassum* along the coastline of Quintana Roo, Mexico using a NN and MODIS data. This study offers a challenge for remote sensing studies providing a capable tool to determine variables allowing to detect *Sargassum* pixel by pixel.

## MATERIALS AND METHODS

948 km of coastline of the state of Quintana Roo, Mexico was defined as the study area to test and to develop the proposed algorithms. Additionally, by using MODIS satellite imagery, a set of data containing official information about zones and dates with and without presence of *Sargassum* was also defined.

### Study area

We selected the entire coast of Quintana Roo, located in the eastern zone of the Yucatan Peninsula, Mexico. One-kilometer-sized MODIS pixels in front of the beach line were selected (from 21.496124 Latitude, -87.546677 Longitude, to 18.477211 Latitude, -88.293625 Longitude), bordering the coast of Quintana Roo. This region is the main vacation destination in Mexico. In addition, the area is located where massive arrivals of *Sargassum* were recorded in 2015 (*van Tussenbroek et al., 2017*) and 2018.

### Dataset definition and processing

To build the dataset, three components were developed: (1) A list of sites and dates with and without *Sargassum* based on official information compiled by the government of Quintana Roo (2018) and field work in 2015 and 2018; (2) sets of Aqua-MODIS imagery, both with and without *Sargassum*, for the coast of Quintana Roo based on the list of dates and sites mentioned above; and (3) Software was developed (extract_data.py) to add the list of sites and images set, which ultimately outputs the data set.

### Area and dates of interest

The list of sites and dates with and without *Sargassum* used in this study was built as follows. First, a pixel list of the entire coastal zone of the state of Quintana Roo at a spatial resolution of 1 km was built. This pixel list included the following parameters: latitude, length, position on the $x$-axis, position on the $y$-axis, township, and date. With the support of Seadas software (https://seadas.gsfc.nasa.gov/), pixels from an AQUA-MODIS image with WGS84 projection were selected. These 948, one-kilometer-sized pixels represented all of the coast of Quintana Roo. Subsequently, based on official information from the state of Quintana Roo (https://www.qroo.gob.mx/noticias/sargazo) and field work, the pixels where *Sargassum* was observed, were labeled. The labeling was done for 29 different dates. A total of 115 different pixels were found with *Sargassum*.

### Selection of Aqua-MODIS swath imagery

Based on the known *Sargassum* arrivals in the coastal zone of Quintana Roo, Aqua-MODIS swath images were also used in the construction of the dataset. The Julian day and the Universal Time Coordinated (UTC time) of all selected swath images (with and without *Sargassum*) were recorded (https://lance-modis.eosdis.nasa.gov/cgi-bin/imagery/realtime.cgi) and then the PDS files (L0) were downloaded from MODIS OceanData (https://oceandata.sci.gsfc.nasa.gov/MODIS-Aqua/L0/). In total, 80 PDS files were downloaded (42 files corresponding with *Sargassum* dates and 38 files without). After the processing and re-projection of all files, an RGB composition for each swath image was made to allow a visual quality check of each image. Due to the presence of clouds in the area of interest, a total of 19 images were discarded (eight images with *Sargassum* and eleven without). Afterwards, 30 files with *Sargassum* and 29 files without *Sargassum* remained as the imagery used in the development of the dataset (*MODIS-Aqua, 2018*). An example of these images is shown in Fig. 1. Not all the images were ideal for network training, because of excessive cloudiness.

### Data processing

Data processing started with the swath images (PDS files) downloaded from the ocean color data website (https://oceandata.sci.gsfc.nasa.gov/MODIS-Aqua/L0/). Based on the software SeaDAS7. 5. 1, PSD file goes through different processing levels to evolve from level 0 (L0) to level 2 (L2). First, L0 file was processed to obtain the level 1A file (L1A). After that, the file GEO was created. Based in files L1A and GEO, level 1B file (L1B) was produced. Next, L2 product was created and then re-projected. Features of MODIS data processing levels can be consulted on the MODIS Nasa website (https://modis.gsfc.nasa.gov/data/dataprod/). Using latitude and longitude pixel-features and the in-house computer program (extract_data.py), surface reflectance (rhos) and top of atmosphere reflectance (rhot) MODIS data corresponding to the coastal zone were extracted and were used to build the dataset. The wavelength bands selected for this study were: 412, 469, 555, 645, 859, 1,240 and 2,130 nm. In the Pseudocode 1 the processing workflow is shown.

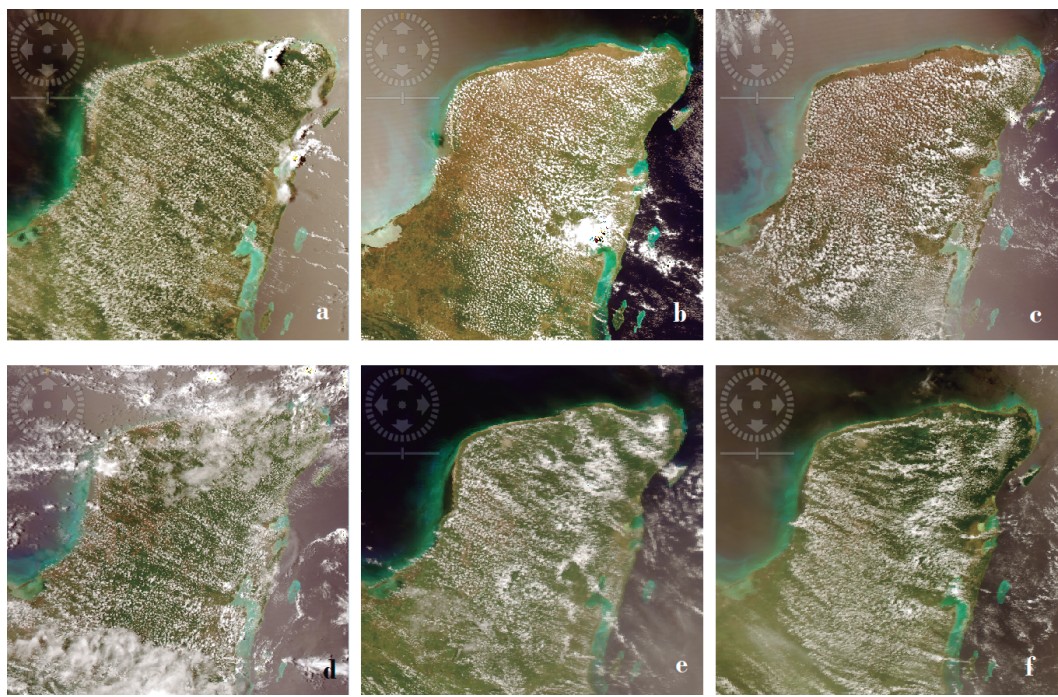

**Figure 1** **Sample of Aqua-MODIS imagery used in this study, (Day/UTC).** (A) From 2015 232/18:55, from 2018: (B) 92/19:20, (C) 94/19:10, (D) 158/19:10, (E) 200/19:45 and (F) 201/18:50.

---

**Pseudocode 1** Scheme of data processing

```
 1:  database ← empty
 2:  for all L0 datafiles do
 3:      L1A ← modis_L1A.py(L0 file)
 4:      GEO ← modis_GEO.py(L1A)
 5:      L1B ← modis_L1B.py(L1A, GEO)
 6:      L2 ← l2gen(L1B)
 7:      Reprojected ← gpt.sh(L2)
 8:      Data ← extract_data.py(Reprojected)
 9:      append_database(Data)
10:  end for
11:  return database
```

---

A database was built with the pixel data of each band for all selected dates. The dataset included 14 different attributes and 4,515 instances, of which 2,306 corresponded to presence of *Sargassum* and 2,209 without. Additional features are shown in Table 1.

## ERISNet a deep learning network for *Sargassum* detection

ERISNet is a deep neural network designed to detect *Sargassum* along the coastline. ERISNet is inspired mainly on two types of architectures; Convolutional Neural Networks (CNN) and Recurrent Neural Networks (RNN). An issue present in virtually all models of Machine

**Table 1  Dataset features.**

| | |
|---|---|
| Number of attributes | 14 |
| Data set characteristics | Multivariate |
| Attribute characteristics | Real |
| Associated tasks | Classification |
| Number of instances | 4515 |
| Number of clases | 2 |
| Number of nistances with *Sargassum* | 2306 |
| Number of instances without *Sargassum* | 2209 |

Learning is overfitting, therefore during the design of the proposed architecture, special attention was paid to maintaining the tradeoff between optimization and generalization of the network by using different mechanisms such as dropout, batch normalization and weight regularization.

The structure of the convolutional block it is formed by four components: Convolutional layer, RELU activation function, Batch Normalization, and Dropout operation. The objective of convolutional blocks is to efficiently extract characteristics or patterns from the input dataset. The main component of the block is a convolutional layer of 1D (dimension one). After conducting numerous tests with different filters and sizes, the decision was made to use a total of [64, 128, 128] filters with a size of [8, 5, 3].

With the aim of avoiding the overfitting, three mechanisms were used: dropout regularization, weight regularization, and batch normalization. Dropout is one of the most effective and most commonly used regularization techniques for neural networks and is used to improve over-fit on neural networks. At each training stage, individual nodes are either dropped out of the net with probability $1-p$ or kept in the net with probability $p$, so that a reduced network is left; incoming and outgoing edges to a dropped-out node are also removed.

Weight regularization is another common way to mitigate overfitting. This involved putting constraints on the complexity of a network by forcing its weights to take only small values, making the distribution of weight values more regular. There are two kinds of weight regularization: L1 and L2 regularization. L2 was used in the convolutional blocks of ERISNet. In L2 (see Eq. (1)) a "squared magnitude" of coefficient as penalty term to the loss function is added.

$$L_2(W) = w_1^2 + w_2^2 + ... + w_n^2 \qquad (1)$$

The convolutional block uses a Batch Normalization operation to increase the network performance. Batch Normalization (BN), is a technique for improving the performance and stability of ANN, providing any layer in a neural network with inputs that are zero mean/unit variance (*Ioffe & Szegedy, 2015*). During the learning process, the type of initialization of weights could cause a digression to gradients, meaning the gradients have to compensate for the outliers, before learning the weights to produce the required outputs. BN regularizes this gradient by normalizing activations throughout the network. It prevents

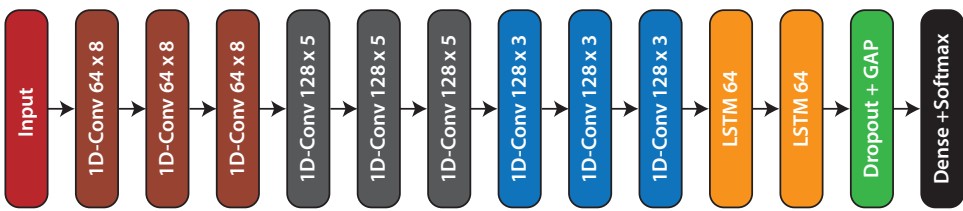

**Figure 2** **ERISNet.** *Sargassum* Deep Neural Network. This architecture is made up of nine one-dimensional convolutional blocks (1D-Conv m × n) with "m" filters whose size is "n", two recurrent blocks (LSTM 64), and finally one Dense (classification) block.

small changes to the parameters from amplifying into larger and suboptimal changes in activations in gradients.

Another component that is part of ERISNet are the recurrent blocks. The main objective of these blocks is to provide memory to ERISNet. Recurrent neural networks (RNN) are a special type of neural network widely used in problems of prediction in time series. Given their design, the RNN allows information to be remembered for long periods and facilitates the task of making future estimates using historical records. Unlike traditional neural networks, LSTM networks have neuron memory blocks that are connected through layers. These memory blocks facilitate the task of remembering values for long or short periods of time. Therefore, the stored value is not replaced iteratively in time, and the gradient term does not tend to disappear when the retro propagation is applied during the training process.

Finally, as in the case of convolutional blocks, recurring blocks also make use of batch normalization to improve network performance. As can be seen in Fig. 2, ERISNet consists mainly of nine convolutional blocks and two recurring blocks.

ERISNet was designed using the programming language Python version 3.7.0 and the library Keras 2.2.4 with TensorFlow 1.10.0 as backend.

TensorFlow is an open source library developed by the Google Brain Team for numerical calculation using data flow graphing programming. The nodes in the graph represent mathematical operations, while the connections or links in the graph represent the multidimensional data sets (tensors). Tensorflow has various automatic learning algorithms and other tools that make it ideal for the development of new methods. Keras is a Python library that provides a clean and simple way to create Deep Learning models on top of other libraries such as TensorFlow, Theano or CNTK.

All the architectures presented in this work were developed and trained using a Lenovo Workstation with Intel Xeon EP processor, 64 GB of RAM, NVidia Quadro K5000 GPU running the Linux operating system Ubuntu 18.04 64 bits.

## RESULTS AND DISCUSSION

Statistical analysis with the information of bands was performed to evaluate the behavior of the current dataset. Next, two algorithms extracted from the literature based on neural networks and machine learning are investigated; these algorithms have shown good results

**Table 2 Mean of the rhos band.** Average values of surface reflectance (rhos), at the different wavelengths (λ), used to this study. Units of the wavelengths are nanometers (nm).

|  | rhos_412 | rhos_469 | rhos_555 | rhos_645 | rhos_859 | rhos_1240 | rhos_2130 |
|---|---|---|---|---|---|---|---|
| **Without *Sargassum*** | 0.131517 | 0.13489 | 0.141123 | 0.124477 | 0.227052 | 0.207291 | 0.085164 |
| **With *Sargassum*** | 0.114489 | 0.12090 | 0.133097 | 0.116607 | **0.247237** | **0.233480** | 0.084166 |

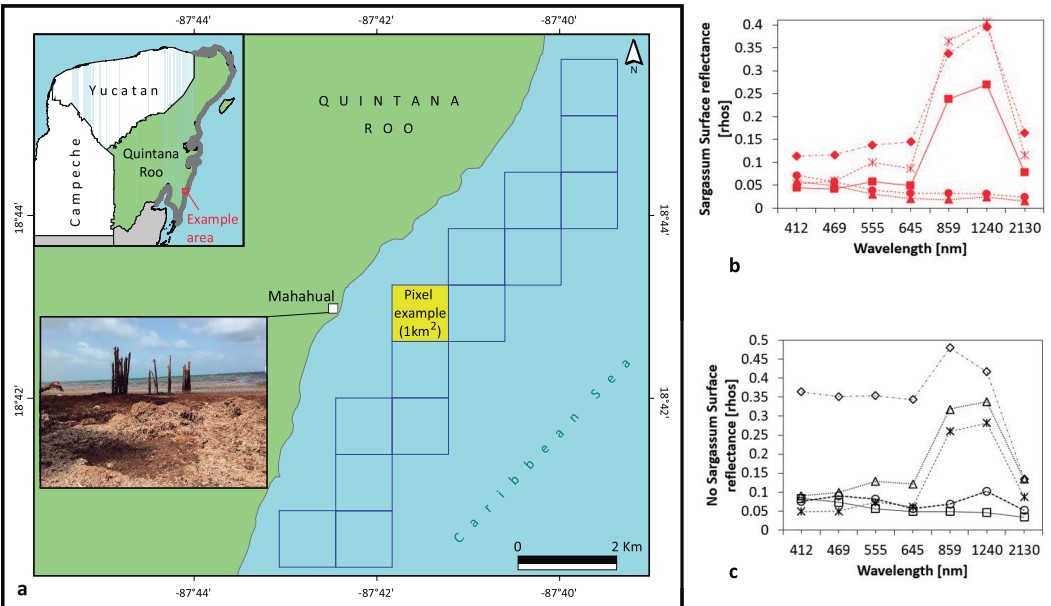

**Figure 3 Study Area: Coastal zone of Quintana Roo.** Represented by 948 pixels of 1 km², close to the beach line. (A) *Sargassum* washed ashore on the coast of Mahahual, Quintana Roo, Mexico on 18 June 2018. For the pixel example AQUA-MODIS rhos values (B) in the case of *Sargassum*; (C) in the case of absence of *Sargassum*. (Figure credit: Holger Weissenberger).

when applied to classification problems similar to that of the present research. Finally, a comparative table is presented with the results of ERISNet and the other competitors.

## Basic statistical analysis

When each of the corrected bands (rhos and rhot) of the generated data with and without *Sargassum* were averaged, small differences were observed. Only the averages corresponding to bands 859 and 1,240 nm were higher in the case of the presence of *Sargassum*. Therefore, a powerful algorithm-tool is needed to efficiently classify the small differences among the values of each pixel and thus classify the presence/absence of *Sargassum*. A basic statistical analysis of data, shows why 859 and 1,240 nm are the bands which FAI index uses. Table 2 shows the means in the case of rhos bands.

In Fig. 3 behavior of rhos values used for a pixel example (Latitude 18.71582, longitude −87.69191), in the presence / absence of *Sargassum* are shown. Trends in Figs. 3A and 3B are similar, and differences are not clear between both cases.

We have chosen a survey (*Wang, Yan & Oates, 2017*) showing a wide comparison between different classification algorithms by using more than 40 different classic datasets. In those studies, authors propose three new classification algorithms based on machine learning techniques and neural networks showing a good accuracy for the different datasets.

In order to compare the performance of the proposed methodology, we have chosen two effective algorithms presented by *Wang, Yan & Oates (2017)*.

## Multilayer perceptron

The multilayer perceptron (MLP) it is defined as the base algorithm for comparison with the rest of the proposals. The MLP used is composed of an input layer, three intermediate or hidden layers, and the output layer. Each of the intermediate layers is composed of 500 neurons that use the rectified linear unit (RELU) as an activation function. To improve the generalization of the neural network, a "dropout" function with values of [0.2 0.2 0.3] respectively has been inserted at the end of each of the intermediate layers. Dropout, applied to a layer, consists of randomly dropping out (setting to zero) a number of output features of the layer during training. Finally, the network has a softmax layer which is widely used in the Multiclass single-layer classification. Formally, each of the blocks of the hidden layers is described as shown in Eq. (3).

$$
\begin{aligned}
\tilde{x} &= f_{(dropout,p)}(x) \\
y &= W.\tilde{x} + b \\
h &= RELU(y).
\end{aligned}
\tag{2}
$$

The MLP was used to perform the classification of the whole dataset (4515 MODIS pixels). The dataset was divided in two groups: a training and test group each with approximately the same amount of data (see Table 2). The learning process was carried out during 3000 epochs, presenting 100 data points in each one (batch size). In Fig. 4A, the result of the learning and test process is shown. On the one hand, the continuous line shows that the MLP has a good degree of optimization (close to 100%), which is to be expected given the learning capacity of this type of network. On the other hand, the dashed line shows the result of the testing process. During the testing process, a set of data which was never used throughout the training process was presented to the MLP in order to see the generalization capacity. As shown, the MLP has a good level of generalization, correctly classifying 83.76% of the test points. There is a wide difference between the optimization and generalization curves, which is usually an indicator of overfitting.

## Fully convolutional network

Convolutional Neuronal Networks have shown a good performance in classification problems. The fundamental difference between a multilayer perceptron and a convolution layer is that MLP layers learn global patterns in their input feature space whereas convolution layers learn local patterns. The basic block of the FCN is composed of a set of filters that are responsible for the extraction of features from the dataset. RELU has been used as an activation function. At the end of the block, the FCN incorporates a

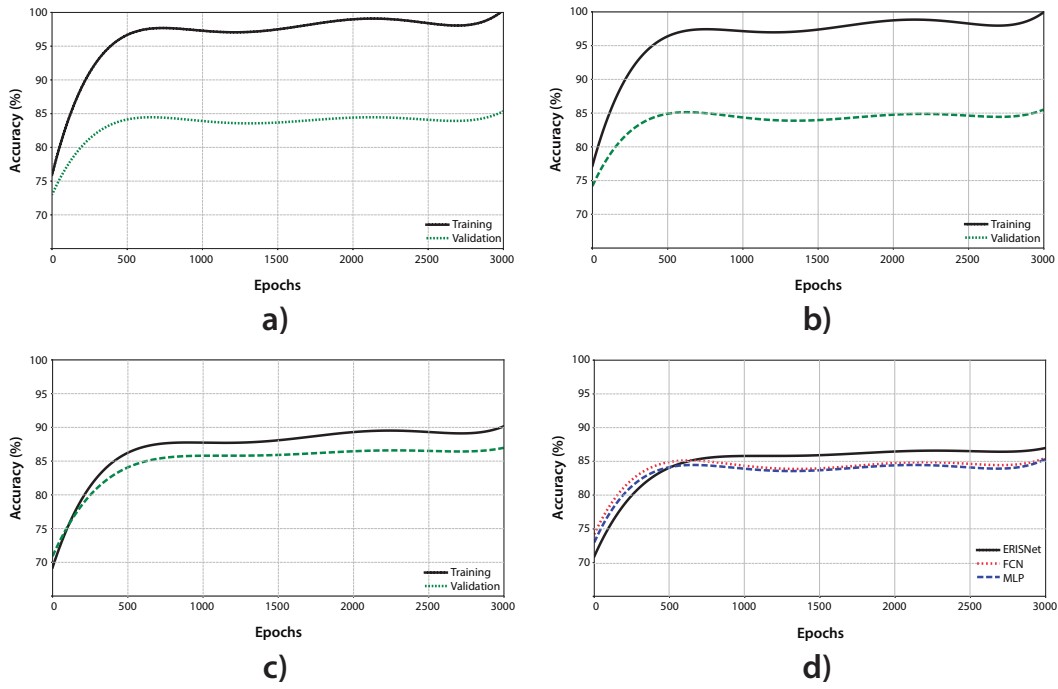

**Figure 4  Results comparison.** This set of plots shows the behavior for multiple neural networks for Sargassum detection during the training process. (A) A multilayer perceptron behavior is shown (MLP); (B) a full convolutional neural network (FCN) behavior also is shown; (C) shows the behavior of ERISNet; (D) depicts a comparison between the three architectures.

new block called Batch normalization (BN). Batch normalization reduces the amount by which the hidden unit values shift around (covariance shift). To increase the stability of a neural network, BN normalizes the output of a previous activation layer by subtracting the batch mean and dividing by the batch standard deviation, getting ten times or more improvement in the training speed.

The FCN used is composed of three convolutional blocks: the first block of the network is composed of 128 filters with eight elements each, the second layer is composed by 256 filters with five elements each, and the last block is composed of 128 filters with three elements each. The objective of this network is to extract from each of the blocks attributes of the data, from general the particular, thus resulting in a good representation of the information contained in the data. With this representation of the data, it is possible to correctly classify information into different classes. Each of the blocks of the hidden layers are formally described by the Eq. (4).

$$
\begin{aligned}
y &= W \otimes x + b \\
s &= BN(y) \\
h &= RELU(s).
\end{aligned}
\tag{3}
$$
Like the MLP, the FCN was used to perform the classification of the whole dataset. With the aim of making a comparison on equal terms, both the dataset and the training parameters used for this model were the same as those presented in the MLP. Figure 4B shows the result of the FCN training and testing process. Similarly to the MLP, the FCN had a fairly high level of optimization and the power of generalization showed good results, correctly classifying the 86.38% of the data points. However, the difference between generalization and optimization suggest the possible presence of overfitting in the network once again.

## ERISNet Validation

At present there are multiple validation methods for neural networks where cross-validation is the most accepted. Cross validation is a statistical method used to estimate the skill of machine learning models. The cross-validation can be divided mainly into two groups: Exhaustive cross-validation and Non exhaustive cross-validation. Among the methods of Exhaustive cross-validation, the following stand out: Leave one out cross-validation (LOOCV), Exhaustive cross validation, Leave out of cross validation, while Non exhaustive cross validation highlights: k-fold cross-validation, Holdout method and Repeated random sub sampling validation.

Due to the characteristics and size of the dataset used, k-fold was chosen ($k = 5$) as the cross-validation method of the ERISNet with $k = 5$. To carry out the cross validation the data set was divided into k parts of which $k - 1$ parts were used as a training set, while the remaining part was used as a validation set. In Eq. (4), the results of the cross-validation are shown. $c_i$ expresses the number of correct classes within the dataset while $e_i$ corresponds to the number of classes correctly classified by the model.

$$MPCE_k = \frac{1}{k} \sum_{i=1}^{k} \frac{e_i}{c_i}. \tag{4}$$

ERISNet was trained, tested, and compared with the rest of its competitors. As in the previous cases, the same criteria were used, that is, the total data set was used by using k-fold cross validation. The algorithm was trained during 3,000 epochs with a batch size of 100 data points, the same seed of random numbers used by MLP and FCN was also used. As can be seen in Fig. 4C, unlike what happened in the case of the MLP and the FCN, the difference between the capacity of optimization and generalization of the network was lower, suggesting that there was no overfitting in the network during the training process. It is important to mention that the level of optimization of the network was less than its competitors, which suggests that if the network is trained during a higher number of epochs this could improve and the generalization could be higher. After the network training, ERISNet obtained a 90.08% success for the classification test points, implying an increase of 7% with respect to the MLP and 4.1% with respect to the FNC.

Figure 4D shows a comparison on the generalization of the MLP, the FCN, and ERISNet. As illustrated, the behaviors of the MLP and the FCN are very similar. However, ERISNet presented an increase in the generalization capacity. As summary, after the dataset classification by using the methodologies mentioned above, the MLP performance was
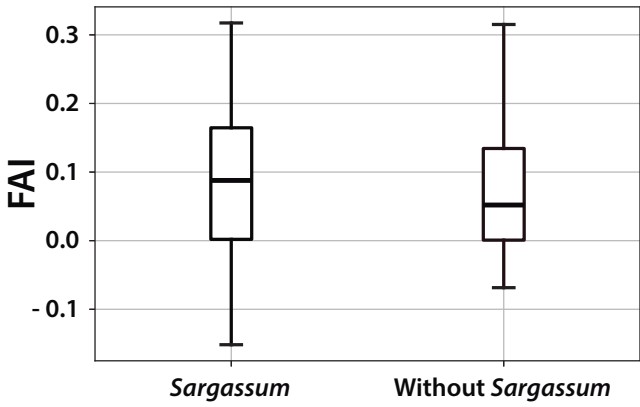

**Figure 5** **FAI boxplot for pixels with and without *Sargassum*.** The boxplot diagram, allows observing the behavior of the dataset used in the present study, from the point of view of FAI index. The median of the pixels in the presence of *Sargassum* is slightly higher than the median of the pixels without *Sargassum*.

83.76% of accuracy followed by the FCN with 86.38%, and the highest accuracy was reached by ERISNet with 90.08%. Based on all the previous tests, it is concluded that the proposed ERISNet algorithm is capable of classifying more precisely new points than other studies.

## FAI index

There are only few studies of *Sargassum* detection along Caribbean coastlines. Therefore, there are no algorithms or methodologies that allow us to make a direct comparison between our proposal and other studies, nevertheless we have calculated the FAI index to the dataset used in the present study in order to know the behavior of our dataset from the point of view of this index. Figure 5 shows the boxplot diagram for the FAI index results. A clear statistical difference can be observed between both datasets. It can be seen that the median of the pixels FAI in the presence of *Sargassum* is slightly higher than the median of the pixels without *Sargassum*. The median of the pixels without *Sargassum* is closer to zero.

It is important to note that 50% of the data with *Sargassum* are less compact than that of those without *Sargassum*, which implies a greater distribution of the values within this 50%. The largest difference appears in the lower whiskers of the *Sargassum* boxplot. This difference indicates that the *Sargassum* FAI values reflect a greater bias with respect to those points without the presence of *Sargassum* (Fig. 5). Table 3 shows the results for the computation of the traditional statistical values made to the FAI index on the data set.

*Sargassum* detection is a complex issue, and for that reason our dataset had to be built with official information of dates, sites and with field work. Our dataset is significative to provide remote sensing information about *Sargassum*, until now not available for this region. Hence, it is possible to apply the dataset in other methodologies or for training other algorithms. Thus to compare the classification performance of ERISNet, two effective algorithms were chosen (*Wang, Yan & Oates, 2017*). As shown in Fig. 4D, ERISNet obtained the best performance.

*Sargassum* detection along the coastline is a challenge for conventional techniques used in remote sensing. The presence/absence of *Sargassum* on satellite imagery along the

**Table 3   FAI index for the dataset.** Central trend statistics values of the FAI index, fed with the same dataset used to train and test ERISNet.

|  | FAI *Sargassum* | FAI without *Sargassum* |
| --- | --- | --- |
| pixels | 2306 | 2209 |
| mean | 0.088595 | 0.072789 |
| std | 0.109495 | 0.081089 |
| min | −0.845114 | −0.845114 |
| 25% | 0.001938 | 0.000794 |
| 50% | 0.087812 | 0.051965 |
| 75% | 0.164528 | 0.134386 |
| max | 0.317351 | 0.315139 |

coastline is not as clear as in the open sea, because there are several transitional ecosystems. Under these conditions the high classification performance of ERISNet allows to observe the small differences between the values of the bands used for the classification and to determine the presence or absence of *Sargassum* in each pixel with a maximum accuracy of 90.08%. This is highlighted for an ANN, since increase 1% unit in the classification requires high performance design and implementation.

Traditionally, detection of suspended matter in open waters is accomplished through satellite products or through well established indexes (*Hu, 2009*; *Hu et al., 2015*; *Hu et al., 2016*; *Cuevas, Uribe-Martínez & Liceaga-Correa, 2018*). Therefore, the present study is innovative, since it has a high classification accuracy of pixels with presence/absence of *Sargassum*, using as input data the corrected bands rhos and rhot. The present research showed that under conditions of high concentration of *Sargassum*, as those presented along the coast of Quintana Roo in 2015 and 2018, it was possible to detect *Sargassum* with MODIS data.

Although the coast of the Mexican Caribbean has high economic and ecological importance, there is no monitoring system that contributes to make decisions facing threats of massive arrival of *Sargassum*. Therefore, the present work is very relevant for this region, because it offers the basis for an early warning system design.

ERISNet could be applied to other coastal areas of the Caribbean region, thereby we propose to design an artificial neural network capable to detect *Sargassum* in open waters and to build new training datasets based on satellite products and well established vegetation indexes.

## CONCLUSIONS

Based on CNN and RNN architectures, ERISNet was developed specifically to detect *Sargassum* along the coastline of Quintana Roo, Mexico.

To our best knowledge, this is the first method using Deep Learning to detect pelagic *Sargassum* along the coastline that considers not only floating but also accumulated *Sargassum*. Based on Aqua-MODIS swath imagery and well-known sites and dates with and without *Sargassum* along the coastline of Quintana Roo, a dataset was built to train and test all the algorithms used in this study. After the learning process, ERISNet achieved a

maximum 90.08% of probability in the classification of pixels with and without *Sargassum*. Additionally, using the dataset for the present study the FAI index was calculated. The measures of central tendency of the FAI index for data with and without *Sargassum* are clearly different. However that index does not offer a percentage efficiency value pixel-by-pixel. Hence, ERISNet goes further as it offers a quantitative value of its own performance.

Several studies have evaluated the threats that coastal ecosystems of southern Quintana Roo have suffered in the last decades (*Alvarez-Filip et al., 2013*; *Hernandez-Arana et al., 2015*; *Arias-González et al., 2017*). However, there is little information that analyzes habitat degradation by *Sargassum*, as it is a relatively new stressor adding to the threats that already exist in the Caribbean. Therefore, an early detection system to alert about massive *Sargassum* arrivals is undoubtedly a challenge for the research of vulnerable coastal zones in the Caribbean, and for the understanding of the threats to these coastal ecosystems.

## ACKNOWLEDGEMENTS

We thank the NASA Ocean Biology Processing Group (OBPG) for providing all MODIS data used in this study. The authors thank Gerald Alexander Islebe and Héctor Hernández-Arana for their invaluable support. We would like to thank Holger Weissenberger for producing Fig. 3, and finally Casey Gibson for her technical support.

### Funding

The authors received no funding for this work.

### Competing Interests

The authors declare there are no competing interests.

### Author Contributions

- Javier Arellano-Verdejo conceived and designed the experiments, performed the experiments, analyzed the data, contributed reagents/materials/analysis tools, prepared figures and/or tables, authored or reviewed drafts of the paper, approved the final draft, software.
- Hugo E. Lazcano-Hernandez conceived and designed the experiments, performed the experiments, analyzed the data, contributed reagents/materials/analysis tools, prepared figures and/or tables, authored or reviewed drafts of the paper, approved the final draft, dataset.
- Nancy Cabanillas-Terán analyzed the data, contributed reagents/materials/analysis tools, prepared figures and/or tables, authored or reviewed drafts of the paper, approved the final draft.

## Data Availability

The raw measurements are available in File S1.

## Supplemental Information

Supplemental information for this article can be found online at http://dx.doi.org/10.7717/peerj.6842#supplemental-information.

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
