# Peer review of "ERISNet: deep neural network for Sargassum detection along the coastline of the Mexican Caribbean"

_PeerJ, doi:10.7717/peerj.6842_

## Round 0.1 · original submission · Major Revisions

I read the manuscript carefully and although it is not my main research field I think the study is interesting to most readers of PeerJ. Whereas I agree with comments provided by both reviewers I disagree with decision of reviewer#2. Certainly, the manuscript lacks some citations but I think it has a strong potential. The application of neural network algorithm I think it provides a novel insight in early warning detection of Sargassum. Probably it is the main strength of the study which needs to be clearly highlighted throughout the ms. Reviewer#2 states that spatial resolution used here is lower than other studies and authors must clearly explain why is not the key of the study.

The manuscript includes too many figures. Be sure that it includes only the necessary ones. Reviewer 2 states that you need more to explain some image treatments, but only include more if you decide to reduce the total number of figures. A maximum of 8-9 figures is usually the most common in a study like this.

Please, provide a point-by-point reply to all issues raised by the reviewer#1 in your revised version.

Reviewer 1 ·

Basic reporting

English and style…

I do not feel qualified to judge English language and style

The authors need to review and add more information on state of the art with respect to the possible effects of such arrivals of Sargassum to the Mexican Caribbean coasts and how the application of the classifier would help the persons in charge of such work and researchers that can make use of this proposal of early warning neural network in decision making.

Professional article structure, figs, tables. Raw data shared. "It ok"

Experimental design

Even though the study is centered in the development of an algorithm for a specific classifying and detecting platform, which is one of the objectives of the algorithm and not only for developers but also for the scientific community, information with respect to the application of the algorithm or in its case the proposal of the neural network is required. Likewise, I don’t see in the manuscript the possible incorporation of other physical and biological variables that could favor distribution and abundance of Sargassum, which could be incorporated in the neural network to develop such early warning that has been proposed or better yet what is the justification for not incorporating them initially.

Validity of the findings

On the other hand, they need to incorporate more information, figures and images, with respect to the analysis of satellite images and their correlation with the specific characteristics of wavelength or specific reflectance of the Sargassum as well as making comparisons of images of the presence and absence of Sargassum. in the Mexican coasts, as well as its respective graph of reflectance values.

Additional comments

I have read the article and find the development of the classifying neural network interesting and novel with its corresponding algorithm to determine the presence/absence of Sargassum in the Mexican Caribbean since its arrival to the coasts has turned out to be a serious problem for different environments both ecologic and economic in the last years. The authors have made a great effort and contribution with the development of this neural network architecture, which shall be of great importance and application.

Annotated reviews are not available for download in order to protect the identity of reviewers who chose to remain anonymous.

Reviewer 2 ·

Basic reporting

I am very alarmed, why the authors did not add antecedents about investigations with the same aim in the Caribbean too. For example: Jean-Philippe Maréchal, Mengqiu Wang, Chuanmin Hu, Gower J.F.R.

Nowadays this research continues but with images with finer spatial resolutions (10 meters) than MODIS images proposed in this article.

In the article the authors said that it is the first work on this topic and its not true.

Example:
https://www.sciencedirect.com/science/article/pii/S2352938516300441#!
https://www.sciencedirect.com/science/article/abs/pii/S0034425716301833#!

The methodology its a good goal but not the general aim.

Experimental design

This is not an original aim. Since 2 years ago there are a really good articles about this research.

They do not mention in their introduction important background on the subject.

Validity of the findings

'no comment'

Additional comments

Congratulation ...A good effort in the methods but there are already research with the same aim with satellite images with finer spatial resolutions and that also quantify the Sargassum.

---

## Round 0.2 · Minor Revisions

Firstly, I appreciate your effort in reviewing your manuscript. It was send again to reviewer#1 which is satisfied with the changes and consider the study improved a lot. I agree with this reviewer. Most of weaknesses and inconsistencies have now been turned into strengths and your study I think is almost ready to be accepted in PeerJ.

Some minor issues need to be considered previous the acceptance; mainly regarding the edition of your ms. I suggest these minor changes:

1) Figures 1 and 4 must be merged. I suggest you use Fig. 4 instead Fig. 1 and then the current Fig. 1 could be removed.

2)Figures 5, 6, 7 and 8 must be merged into one with the four graphs. Something like this arrangement:
5 / 6
7 / 8

3) Table 3 is unnecessary; it should be removed and results commented in the text.

4) Results and discussion should be merged in one single section. In the current version, the discussion section alone does not contribute much. However, I think that since the study is clearly methodological, if you incorporate the discussion content in the results section, the manuscript gains much in clarity.

Once these changes have been made, you have the certainty that your article will be accepted.

Best regards,

Salva

Reviewer 1 ·

Basic reporting

The article is adjusted to the professional standards, the information included in the introduction and background are sufficient to demonstrate how the work fits the area of knowledge.

Experimental design

The research carried out by the authors is original and of high impact for the study area, and the information of the method contained in the document is relevant and significant to be followed up and can be replicated by other researchers and provide new knowledge or new methodologies for the study of sargassum.

Validity of the findings

The results are good and statistically proven, which leads to well-established conclusions by answering the main question.

Additional comments

Dear authors. The work done in the comments and corrections substantially improved the content and sense of the research, so I have no problem in recommending its publication. Good job.

Annotated reviews are not available for download in order to protect the identity of reviewers who chose to remain anonymous.

---

## Round 0.3 · accepted · Accept

Congratulations! Good job!